# A Spatial Analysis of the Voting Patterns in the South Korean General Elections of 2016

**Hyun-Chool Lee** [1] and **Alexandre Repkine** [2],*

1   Department of Political Science, Konkuk University, Seoul 05029, Korea
2   Department of Economics, Konkuk University, Seoul 05029, Korea
*   Correspondence: repkine@konkuk.ac.kr

**Abstract:** In this study we analyze the spatial patterns in the South Korean voting behavior in the context of the 2016 general election along with the socio-economic determinants of the South Korean voters' choice. To this end we applied spatial econometric analysis to a unique dataset on the outcomes of the 2016 general elections in South Korea at a highly disaggregate level of 229 provinces. Our empirical model accounts for three types of spatial dependence in the data that has to do with the fact that geographic proximity may imply similar voting behavior. Our empirical findings align well with the existing evidence on South Korean voting behavior, in particular regarding the influence produced by the voters' region of origin, and their age. Surprisingly, we do not find economic characteristics such as the regional income per capita or the rate of unemployment to produce a statistically significant effect on South Korean voters' choice. However, our results imply that a sound fiscal policy by the local government may act as a signaling device distinguishing between a conservative and a liberal political agenda. Our finding of the older voters leaning towards the conservative edge of the political spectrum suggests that the "silver democracy" now actively discussed in the South Korean media is increasingly assuming more conservative traits.

**Keywords:** South Korean general election; silver democracy; population ageing; regionalism





## 1. Introduction

The First Law of Geography formulated by Tobler (1970) says that "Near things are more related than distant things". This study uses spatial econometric estimation methods in order to detect and analyze possible regionalism in the general election of 2016 in South Korea (henceforth Korea). We provide strong empirical evidence in support of the regional voting patterns present in this election and estimate the effects of an array of socioeconomic characteristics on the Korean voter choice.

In the Korean context regionalism is mostly defined as a political conflict between Korea's South-Western provinces known as Honam and her South-Eastern provinces known as Yeongnam. While Korean voting behavior has long been known to be influenced by the region of origin of political parties' leaders after the event of democratization that had occurred in 1987, e.g., Sonn (2007), little effort has been made to evaluate the extent of this regionalism or its determinants in a spatial econometric framework that allows separation of the effects of the socioeconomic determinants from the influence of the neighboring areas.

We posit the existence of at least three major sources that may drive regional voting differences in principle, namely, interregional differences in the level of socioeconomic development, the historical legacy making voters in, e.g., Honam support a political party whose stronghold is also in Honam, and the influence of the surrounding regions. We discuss these factors in detail in Section 2. While the effect of the first two factors can be captured within a conventional multivariate regression framework, the third effect is only possible to analyze via spatial econometric methodologies, whose use is also strongly supported by the statistical tests we conducted on our data.



*1.1. Regionalism in Korean Electoral Politics as a Potential Threat to Democratic Rule*

Regional voting patterns have been present in Korean voting patterns in different forms ever since May 1954, shortly after Korea's Liberation, when the third general election was held. Prior to 1987 when the country was swept by a number of demonstrations calling for a democratic change, Korean electoral politics were dominated by a rural-urban division in the sense that the ruling party would traditionally find support in the rural areas while the opposition's stronghold would be in the urban areas. In this sense, the nature of Korean regionalism prior to 1987 was rather different from the regionalism existing today.

After 1987, however, it is the conflict between Honam and Yeongnam that became a defining characteristic of Korean regionalism as it is known today in the sense that the voters in either region would provide an overwhelming support to one particular party or a candidate representing it. For instance, as we demonstrate in Section 3.3, in the 17th presidential election the two Honam provinces voted almost entirely in favor of the opposition party, while the two Yeongnam provinces overwhelmingly voted for the ruling party. We identify similar voting patterns in the general election of 2016 studied in this paper. Some authors, e.g., Lee (2003) argue that the region of voters' origin was the most important characteristic affecting Korean voter choice after 1987 with the difference between Honam and Yeongnam voters being especially salient. We provide a detailed historical review of the regional voting issues in Korea in Section 3.

Similarly to Cho (1998), we believe that regional voting behavior can potentially undermine democratic rule in Korea since it discourages competition between the political parties' socio-economic agendas by replacing it with a conflict of regional sentiments. Indeed, in an environment that makes it possible for a particular political party to gain a victory in a certain region the political parties' socio-economic agendas are likely to be replaced by regional sentiment that the politicians will be eager to exploit and enhance, thus creating a vicious circle. In fact, in a homogeneous country like Korea that is devoid of race, ethnicity or religion-based cleavages, mobilizing the sense of belonging to a certain region among voters appears to be an easier way of gaining voter support compared to developing a distinct socio-economic policy agenda. Some studies, e.g., Yoon (2017) go as far as to conclude that the tendency of Korea's Honam and Yeongnam regions to vote in a uniform fashion is mostly a function of the voters' psychology rather than of the political or economic agendas pursued by the political parties they are supporting.

A vast array of socio-economic determinants have been identified in the literature on election outcomes, as discussed in the literature overview in Section 2.1. The purpose of this study is to evaluate the relative importance of the voters' region per se versus those regional characteristics that are related to the performance of the local governments such as, e.g., the rate of unemployment or the regional level of per-capita income. The most relevant theory in this regard appears to be that of the economic disparities between Honam and Yeongnam. Studies such as that of Kang (2011b), among others, argue that president Park Chung-Hee's industrialization policies conducted in the 1970s favored his native region of Yeongnam at the expense of Honam, contributing to the feelings of animosity between the two regions. In this case the reluctance of Honam voters to support any political party or candidate coming from Yeongnam falls well within the predictions of the reward-punishment model that says voters will reward those political forces that are likely to improve their economic conditions, see, e.g., Tillman and Park (2009).

Whether Korean regionalism is a function of the regional economic environment, or that of the sense of belonging to a particular region per se is an important question to answer as it affects our understanding of how detrimental Korean regionalism potentially is to Korea's democratic rule. If it is regional sentiment alone that mostly determines the results of the elections in Yeongnam and Honam, the competition between political parties' political and socio-economic agendas plays a secondary role, and democratic rule is undermined.

### 1.2. How Important Is a Region per se? The Value of Spatial Econometric Analysis

The term "spatial econometric analysis" can be misleading as it is used in several contexts in the literature. Thus, in the seminal studies (Poole and Rosenthal 1984) of the US presidential elections for the period between 1968 and 1980, and that of Dow (2001) that compare the majoritarian and proportional election results in the four Western economies, the term "spatial" refers to the multi-dimensional ideological space based on the general idea that voters will support that political actor who is the closest to them, e.g., in the sense of the Euclidean metric, ideologically. In this study we use the term "space" in its geographical sense so that proximity is understood in terms of the geographical distance of the voters in different regions to each other.

The abovementioned Tobler's law implies that voters in geographically adjacent regions will vote similarly. If voter choice in the neighboring regions is an important factor affecting Korean voters' behavior, capturing regional influence with, e.g., the regional dummies in the OLS framework will confound the effect of the region per se with that of the adjacent constituencies. The value of spatial econometric analysis, (see Luc Anselin (1988) for a classical reference), is that it allows one to separate the effects of the neighboring regions from the "pure" effect of the region per se. Technically this is done by adding to the set of explanatory variables a weighted average of the voting shares in the other regions, where the weights represent the extent of geographical proximity. Intuitively, the "pure" regional effect is captured by the regional dummies after the regional proximity has been taken into account by the spatial econometric modeling. Statistical significance of these regional dummies can be interpreted in terms of importance of the voters' region per se, i.e., in terms of the strength of the regional sentiment alone that in turn may be the result of political mobilization, see (Mori 2012, pp. 33–42). We discuss the implications of failing to account for the influence of the neighboring regions by, e.g., applying an OLS estimation framework in Section 5 and demonstrate that the OLS estimation in case spatial spillover effects are present can be seriously biased both in terms of the magnitude of the estimated coefficients, and in terms of their statistical significance.

Spatial econometric analysis of election outcomes has gained in popularity in recent years. Recent contributions include the spatial analysis of the US county-level presidential election outcomes in (Kim et al. 2003), the application of the spatial econometric approach to the 2017 regional election outcomes in Catalonia in (Maza et al. 2019), or the analysis of the 2010 British general election in (Cutts et al. 2014). In the Korean context, a recent contribution (Lee and Repkine 2020) demonstrates the importance of applying spatial econometrics to the analysis of the 2017 presidential elections in Korea. To our knowledge, this is the only study of electoral outcomes in Korea based on spatial econometrics. In this study we are extending the analytical framework developed in (Lee and Repkine 2020) to the Korean general elections of 2016, which allows us to underscore the importance of accounting for the interregional spatial spillovers in the study of Korean regionalism.

### 1.3. Tobler's Law, Social Sciences, and Spatial Econometrics

While Tobler's law, formulated more than fifty years ago, makes intuitive sense, debates around it continue to this date. Some authors went so far as to call it an "unlawful relation and a verbal inflation", see (Smith 2004). Others such as (Flyvbjerg 2001) argue that social sciences are incapable of producing laws in principle, one of the reasons being that once an empirical regularity is made known, or discovered, it will be altered by human behavior, thus invalidating the regularity itself. Similarly, Giddens (1982) questions the usage of the term "law" in social sciences. Yet, while probably not as precise as, e.g., Newton's law of gravity, laws are not only present in social sciences, but are widely recognized as such. Well-known examples include, for instance, Pareto's law of income distribution, and the law of diminishing marginal utility. While the debate on whether it is appropriate to call Tobler's observation a law is still going on, the idea that the values of certain attributes of a specific location are a function of these attributes' values in the

neighboring location has long become basic in the widely applied spatial econometrics framework, see (Luc Anselin 1988), that we are applying in this study.

Despite the ongoing debate regarding the validity of qualifying Tobler's statement as a law, or the issue of technological progress rendering the influence of geographical distance less relevant to the analysis of spatial interactions, the ideas of spatial dependency, such as spatial autocorrelation or spatial lag, see (Anselin 1995), are currently firmly rooted in the analysis of human behavior within a geographical context. As Rushton (1993) puts it, "Since all behavior occur in spatial contexts . . . the spatial arrangement of things usually affects behaviors. The core of spatial analysis to the behavioral geographers is the analysis of behavior in its spatial context."

One reason why "the spatial arrangement of things" might affect voters' behavior is that, as posited in (Braha and De Aguiar 2017), voters tend to imitate each other, with the strength of this tendency being a function of geographical closeness. The authors apply their theory to a wealth of a century-long data on the US presidential elections at a county level to analyze the electoral outcomes within a framework of spatial statistical analysis. An important finding of their study is that voters' proclivity to imitate other voters is spatially clustered, implying the presence of spatial autocorrelation in the neighboring regions. Whether this spatial clustering is due to the processes of ideological contagion, herd behavior, or a particular network relationship between passive and opinion leaders, is a subject of ongoing research that will set a solid theoretical foundation behind Tobler's insights.

Regionalism in Korean voters' behavior defined by (Stockton and Heo 2004) as the proclivity of certain regions to vote for a specific political party or a politician coming from a specific region is a well-recognized phenomenon. Regionalism can also be defined as that particular kind of spatial distribution of voters whereby a particular political party gets most votes in the counties belonging to a specific macro-region such as, e.g., a province. Since we find evidence discussed in Section 5.1 that Korean voters residing in the country's South-East (Yeongnam) and South-West (Honam) tend to vote predominantly for the same political party, the particular spatial distribution of Korean voters we are documenting in this study falls within the definition of regionalism.

If voting shares obtained by a political party in a specific county are affected by the voting shares of the same party in the nearby regions, one cannot evaluate the effects of voter characteristics and the associated socioeconomic determinants on voter choice without specifically accounting for voter choice in the neighboring regions. As discussed in (Maza et al. 2019), such an estimation will be biased and thus unreliable. Spatial econometrics was developed exactly in order to deal with this kind of bias, which is why we are adopting the spatial econometric framework in this study. In other words, recognizing the presence of Tobler's law in Korean elections necessitates the type of analysis that takes voters' spatial distribution into account.

This paper is organized as follows. In Section 2 we discuss the rationale behind socio-economic determinants of the voting shares we are using in our analysis, discuss the main feature of Korean regionalism, and provide a rationale for employing spatial econometric analysis. In Section 3 we provide a detailed historical review of Korean regional issues and discuss the main features of the 2016 Korean general elections. We discuss our dataset and the estimation methods in Section 4. Section 5 presents empirical results. Section 6 provides discussion, and Section 7 concludes.

## 2. Literature Review

### 2.1. Socio-Economic Determinants of Korean Voter Choice

Regional GDP per capita, or RGDPC, appears to be an important determinant of the voting outcomes in Korea since, as noted by, e.g., (Jung 2016), wealthy Korean voters tend to be more conservative. On the other hand, (Inglehart and Norris 2000) argue that more affluence results in increased aspirations for democracy, which makes evidence of the effect of RGDPC on the voting outcome a matter of empirical investigation. Thus, an affluent

voter may well choose to support a pro-reform, progressive political party rather than a conservative one. In Korea, a group of affluent voters who are nevertheless reform-oriented are called Kangnam-left, where Kangnam refers to a district in Seoul characterized by very expensive housing prices. One interesting theory put forward by (Inglehart and Norris 2000) is that more affluence results in more aspiration for democracy as a non-monetary value. In the Korean context, this is analyzed by (Kang 2011a).

The rate of unemployment U has also been found to be an important factor affecting the voting behavior in numerous studies such as those of (Park and Reeves 2020) or (Caleiro and Guerreiro 2005). We also include the share of older voters OLDSH where the latter are defined as those over the age of 65. Older voters have been shown to lean towards the more conservative edge of the ideological spectrum. For instance, Mori (2012) notices that in the presidential election of 2002 younger voters favored reform-oriented Roh Moo-Hyun, while the older voters mostly voted for his conservative rival, Lee Hoi-Chang. The threshold of sixty-five years was chosen since this is the age upon reaching which one is entitled to social welfare payments such as, e.g., the basic pension and long-term care insurance.

An important historical legacy in the Korean context that must be accounted for is the urban-rural divide that defined ideological cleavages in Korean society prior to the democratization of 1987, see (Lee 1998). The urban-rural divide in Korea commonly refers to the tendency of rural voters to support the ruling party with the urban voters favoring the opposition. One explanation is that voters in the urban areas tend to be more educated and are striving more for democratic values compared to their rural counterparts, as discussed in (Kim 1995). At the same time, rural voters respond more to the material stimuli provided by the ruling party during the election campaign compared to their urban counterparts since incomes in the rural areas are generally lower, see (Yoon 1989). To account for the possible influence of the urban-rural divide we include regional employment shares in the manufacturing (MSH) and the wholesale and retail trade (SERVSH).

We believe it is important to include the labor-market characteristics, namely the labor force participation LPART that captures the effect of the extent to which a region's population is economically active, and the regional unemployment rate U that can be roughly interpreted as the probability of not finding a job if one is eligible to work and is actively looking for a job. The inclusion of these two determinants into the list of the socioeconomic determinants of Korean voter choice makes sense in light of the reward-punishment model of voting behavior discussed, e.g., in (Tillman and Park 2009). This model implies that those political forces that are perceived to be improving the economic well-being of voters in their constituencies, are likely to get more votes. This theory also warrants the inclusion of regional GDP per capita (RGDPC) discussed above.

We are also controlling for the regional budget balance BUDBAL in order to control for the quality of the regional governments' performance. The importance of this variable lies in the fact that it accounts for the amount of spending on welfare and social overhead capital by a local government. Existing studies find that the value of the local budget balance affects voting behavior. Thus, studies like (Happy 1992) and (Lowry et al. 1998) conducted at the level of municipalities in Canada and the United States, respectively, argue that a positive local budget balance increases the probability of a local mayor being re-elected. Some studies such as e.g., (Kim and Kim 2011) in the context of Korean local elections argue that the sign of the relationship between budget balance of the local government and voting shares depends on the strength of the political competition, with stronger competition increasing the voting share of a local government running sound fiscal policies.

We include the share of high-school graduates in total regional population (GOJOLSH) since, as shown in (Lee 2007), more education is conducive to being more reform-oriented in Korea. Since voting choice is a personal matter, we assume it may also be influenced by the voters' family background, which is why our list of voting shares' determinants includes the share of single-member households (SINGSH), and the share of households with three or more children (DASANSH). Both types of households are likely to be economically

disadvantaged, which warrants their inclusion in the list of the socioeconomic determinants according to the reward-punishment model, see (Tillman and Park 2009).

*2.2. Historical Legacy of Korean Regionalism*

We include two regional dummies, namely, Yeongnam and Honam, in order to account for Korean regionalism. Yeongnam consists of Korea's two South-Eastern provinces, while Honam comprises two provinces in the South-West. Using the definition in (Stockton and Heo 2004), regionalism refers to the tendency of higher voting shares for a particular political party or a candidate to be concentrated in a certain region. Lee in (Lee 2003) analyzes the outcomes of Korea's 2002 presidential election and finds that the voters' region turned out to be the most important determinant of Koreans' voter choice. Similarly, Lee and Repkine (2020) find evidence of Korean regionalism in the context of Korea's presidential election in 2017. Our empirical results in this study also demonstrate the strong influence of Korean voters' region of origin on their voting behavior.

Korean regionalism is rather surprising since, being an ethnically homogeneous state, Korea lacks cleavages along racial, ethnic, or religious lines. Mori (2012) offers several explanations behind Korean regionalism. The historical line of argument is based on the fact that the modern-day Honam is located within the historical area of the ancient Baekje kingdom, while Yeongnam is in this way a "successor" to another ancient kingdom, Shilla. Historically, the two kingdoms were rivals. However, studies like (Kim 2003) argue that, since there was little evidence of any cleavages between Honam and Yeongnam by the end of the Korean war, historical legacy can hardly be an explanation for today's regionalism in Korea.

Kang (2001) argues that Korean regionalism is likely to do with the fact that Yeongnam is home to Korea's controversial autocratic president Park Chung-Hee, while his democratic rival Kim Dae-Jung comes from Honam. Ideological differences are not likely to have been at the root of the conflict between Honam and Yeongnam; however, President Park's economic policies favored the development of Yeongnam in the 1960s and 1970s, see (Mori 2012). Another important development is that between 1961 and 1997, when every Korean president originated from Yeongnam, making scholars such as Sonn (2007) claim that the presidents' regional origin produced a serious bias in the political elites' recruitment.

## 3. An Overview of the Regional Issues in Korea

*3.1. Between Liberation and the Outbreak of the Korean War*

There have been a total of 21 general elections in Korea since the first such election took place on the 10 May 1948, after Korea's liberation from Japanese rule. The first two general elections were held before the breakout of the Korean War on June 25 June 1950. With political parties virtually non-existent in Korea at that time in the strict sense of the word and the general public having had no experience with the election process, it is not surprising that as a result of the first regional election the National Assembly was dominated by independent candidates who acquired 42.5% of the total seats. In this context, the issue of regionalism can hardly be said to have been relevant. The second election conducted in May 1950 was similarly dominated by independent candidates who acquired 60.5% of the total seats in the National Assembly.

*3.2. Rural-Urban Divide before the Democratization of 1987*

The first signs of regionalism can probably be seen in the third general election (then for the lower house) held in May 1954 just after the Korean War had ended on 27 July 1953. Despite the fact that this election was hardly legitimate with the ballot boxes rigged and many opposition candidates having been refused registration, two of Korea's South-Western provinces, known as Honam, namely, North and South Jeolla, displayed a markedly different voting behavior compared to the rest of the country. Not only was the Honam voters' support rate for the ruling Liberal Party the lowest at 29% compared to, e.g., 36.5% in Yeongnam and 55.5% in Gangwon-do, the difference in support rates for the Liberal

Party and its major competitor the Democratic National Party was the lowest of all regions at 14.8 percentage points, whereas in the other regions this difference was at or in excess of the level of 30 percentage points, see (Mori 2012).

Prior to a series of demonstrations that swept through Korea in June 1987 because of the then President Chun Doo-Hwan's decision to oppose the direct presidential election, Korean regional and presidential elections were dominated by the so-called *yeo-chon-ya-do* paradigm that can be loosely translated as "ruling party-rural areas, opposition party-urban areas" in a total of twelve general elections conducted prior to 1987. Within this paradigm the ruling party tends to find most of its support in the rural areas with the cities being the stronghold of the opposition.

Thus, in the fourth general election held in May 1958 the ruling Liberal Party won in all regions except for the Gyeonggi province, while it lost in the urban areas. Similar patterns are observed in virtually all National Assembly elections held prior to 1987 with most differences in voter choice observed between rural and urban areas, geography playing a rather simple role.

In Figure 1 below we present voting shares obtained by the ruling Democratic Republican Party and the opposition Civil Rights Party in the sixth general election. In Pane A where we plot voting shares for the six major provinces in Korea, we observe that the ruling party wins everywhere by a substantial margin except for the Gyeonggi province that surrounds Seoul. In particular, we do not observe any outstanding difference between the regions of Honam and Yeongnam, even if in the former the ruling party wins by a smaller margin compared to the latter. The island of Jeju appears to have shown overwhelming support for the ruling party for some reason. However, regionalism viewed in terms of certain regions supporting particular political parties does not appear to be supported by the data, which is in line with what most scholars would say about the Korean elections held prior to 1987, see (Lee 1998) and (Cho 1994). Thus, big cities such as, e.g., Seoul or Busan voted predominantly in favor of the opposition party irrespectively of these cities' geographical location. Similarly, the provincial and rural regions (*si* and *gun*) have supported the ruling party by a substantial margin no matter what region they belonged to, which is in line with the *yeo-chon-ya-do* paradigm discussed earlier.

**(A)**    **(B)**

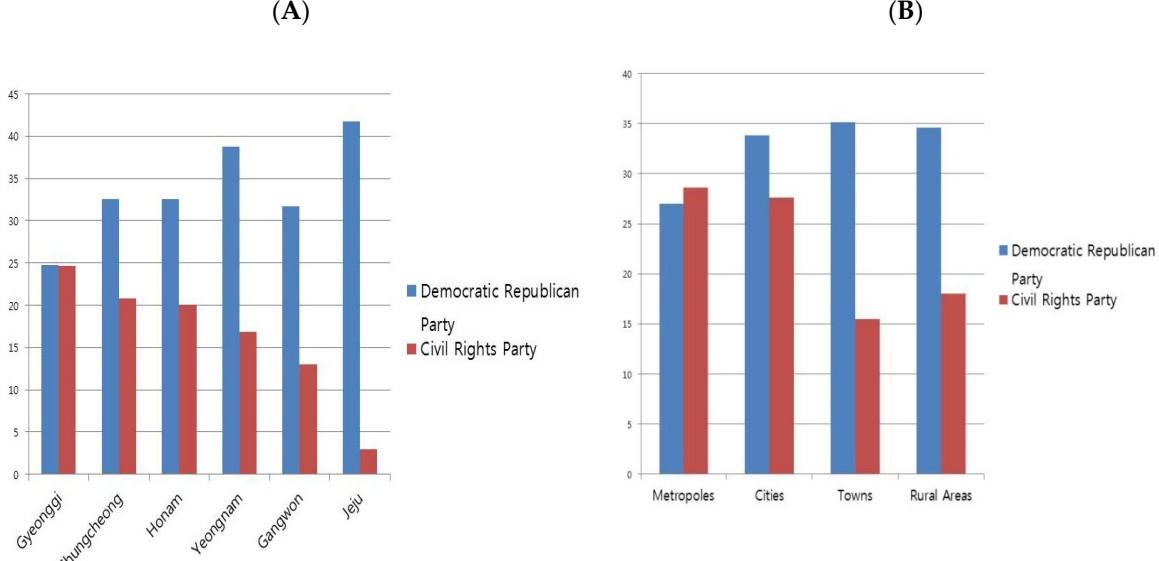

**Figure 1.** Voting Shares in the Sixth General Election, November 1963. (**A**) Voting Shares by Province. (**B**) Voting Shares by Urban/Rural Division.

*3.3. Regional Cleavages between Korea's South-West and South-East after 1987*

Most scholars such as (Sonn 2007) agree that the democratic transition of 1987 resulted in a major shift in Korean voters' behavior that increasingly started to reveal regional

cleavages in the sense that, in certain regions, the winning political party would gain a sweeping, rather than a marginal, victory over its competitors. This phenomenon, that is commonly referred to as regionalism, has been particularly pronounced in the Korean regions of Honam and Yeongnam. The former consists of two provinces, North and South Jeolla, while the latter is comprised of North and South Gyeongsang. Given the centuries-old historical conflict between Honam and Yeongnam, it would not be an exaggeration to call the voter choice differences between these two regions the defining characteristic of Korean regionalism, even if studies such as that of (Dormel and Hassink 2010) contend the validity of a history-based explanation of Korean regionalism. Lee and Repkine (2020) draw on the discussion in (Mori 2012) to list several reasons for the observed regional patterns in Korean voters' behavior.

In Figure 2 we present voting shares observed in the 17th general election in Korea. It is striking how the Honam provinces almost entirely support the opposition Uri Party. In contrast, the Yeongnam provinces vote in favor of the ruling Grand National Party with the winning margin especially impressive in the North Gyeongsang province. In Pane B we report voting shares obtained in Korea's major cities. Seoul votes similarly to the Gyeonggi province that surrounds it in the sense of no substantial winning margin for the opposition Uri Party. The city of Gwangju located in Honam votes in unison with the Honam provinces of North and South Jeolla, while in the city of Busan that belongs to Yeongnam the ruling Grand National Party wins by a similar margin as in the Yeongnam regions of North and South Gyeongsang. Similar observations can be made for the Korean general elections held after 1987 with the urban-rural divide almost vanishing and the regional differences, especially those between the regions of Honam and Yeongnam, becoming increasingly more pronounced.

**(A)** **(B)**

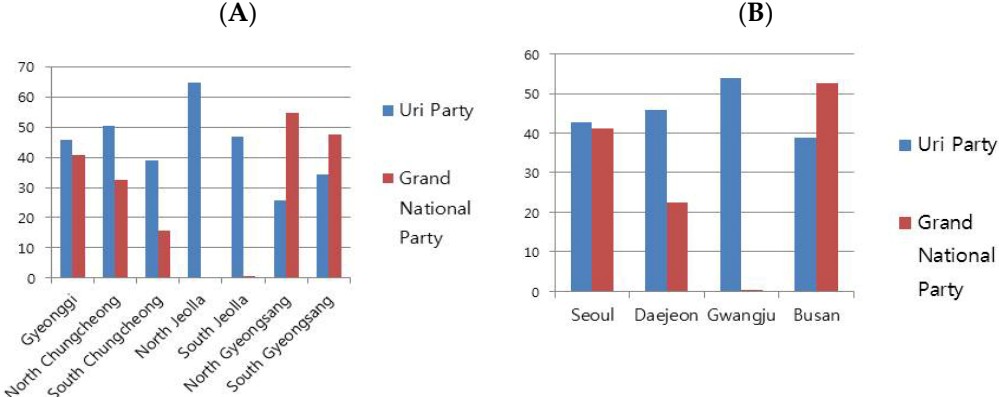

**Figure 2.** Voting Shares in the Seventeenth General Election of 2004. (**A**) Voting Shares by Province. (**B**) Voting Shares by Urban/Rural Division.

While differences between Honam and Yeongnam are the essence of Korean regionalism, voters' preferences in Jeju island appear to be rather different, as well from those in the rest of the peninsula, albeit to a smaller extent. Thus, we have already seen that the Jeju island voters provided the largest winning margin for the ruling party in the sixth general election. Similarly, in the ninth general election the Jeju island voters supported the ruling party by a 27% margin, by far the largest among all Korean regions. One of the factors probably contributing to this specific voting behavior in the Jeju island is the fact that the Jeju population speaks a distinct Korean dialect that is often mutually unintelligible with either standard Korean or its various dialects spoken in the peninsula proper.

It is most likely due to this cultural idiosyncrasy that Jeju region has a culture of election behavior that is different from the other regions, and because the Jeju region is the only region in Korea that does not have a land border with the other regions, which results in a strong tendency to be emotionally drawn to local family and school ties. As a result, political support in Jeju was mostly based on personal networks rather than the political

parties' policies. Thus, even if prior to the event of democratization the Jeju voters tended to support the ruling political party, after 1987 these voters' preferences shifted toward supporting the independent candidates.

Finally, the North and South Chungcheong provinces also tend to display a specific voting pattern. For example, in the 13th general election, the Chungcheong provinces voted almost uniformly for the New Democratic Republican Party, a political party headed by Kim Jong-Phil who hailed from these provinces. In the 16th general election, the North and South Chungcheong provinces also overwhelmingly supported the United Liberal Democratic Party, a pattern not observed in the other Korean regions.

To summarize, while the urban-rural divide known as *yeo-chon-ya-do* in Korea used to shape general elections in the country prior to 1987 when democratization took place, it is the regional differences that appear to have been a major driving factor between Korean voters' choice after that year. Although the differences between Honam and Yeongnam regions are definitely the defining characteristic of Korean regionalism, the island of Jeju and the North and South Chungcheong provinces exhibited a special voting behavior more than once. In our analysis we feel, therefore, that it is necessary to control for the regional differences between Honam, Yeongnam, Jeju island and the two Chungcheong provinces, while at the same time accounting for the urban-rural divide in case the latter has preserved any residual influence.

### 3.4. An Overview of the Korean General Election of 2016

Three major political parties competed for seats in the Korean National Assembly on the 13 April 2016, namely, the conservative ruling Saenuri Party, the Democratic Party representing Korea's liberal opposition, and a centrist People's Party that split from the Democratic Party a few months before the election. The fourth competitor, a left-wing progressive Justice Party, ended up garnering a 5.7% support rate on average, which is significantly less than an average 25.16% support rate obtained by the People's Party, its nearest competitor.

The 2016 general election in Korea was a tight race with the opposition Democratic Party defeating the conservative ruling Saenuri Party by a single seat in the National Assembly. In the national votes for party lists, however, the Democratic Party came third gaining an average of 25.5% in proportional representation (PR) of the votes in Korea's 229 voting constituencies. In contrast, the ruling Saenuri Party was able to obtain 33.5% in the PR votes on average, with the People's Party coming in second with a PR voting share average of 26.7%. Finally, the Justice Party was only able to garner 7.2% of the PR votes, which is why we will limit our analysis to the three major competitors, namely, the Democratic, Saenuri, and the People's parties.

In Figure 3 we display voting shares' quantile distributions for the three major competitors in Korea's 2016 general election in 229 regions. Higher voting shares are represented by the darker regions.

Korea's South-Eastern Yeongnam regions predominantly voted for the ruling Saenuri Party, while its two competitors gained most support from the South-Western Honam regions. This pattern is in line with the widely recognized issue of Korean regionalism discussed in Section 2.2 underscoring the extent to which voting behaviors in Honam and Yeongnam are different. It is interesting how in the Seoul area the voting pattern does not appear to be biased towards any political party in particular which can be explained by the fact that Seoul's population consists of natives of virtually all Korean regions.

| Saenuri Party | Democratic Party | People's Party |
|---|---|---|

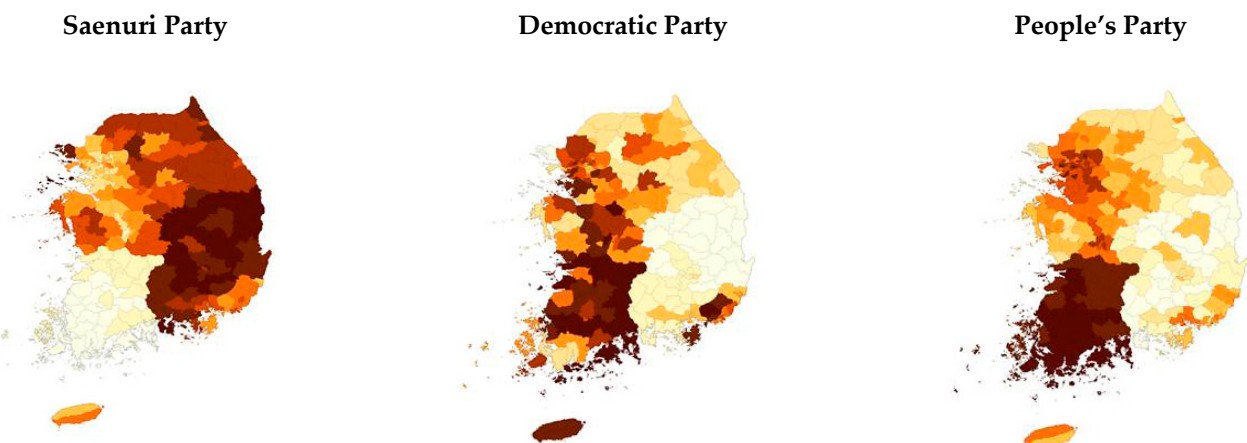

**Figure 3.** Regional Distribution of Voting Shares for the Saenuri, Democratic, and People's Parties in Korea's General Election of 2016.

The regional voting patterns illustrated by Figure 3 may be driven by a number of factors. First, socio-economic characteristics appear to be important determinants of voter choice as implied by the reward-punishment model, see, e.g., (Tillman and Park 2009). This model predicts that the voters will reward the candidates of those parties whose rule is associated with better economic outcomes. Second, Korean regionalism is likely to be a powerful factor influencing the election outcome. Indeed, as suggested by Figure 1, Korea's South-East votes oppositely to Korea's South-West. Finally, voting behavior in any one county is likely to be a function of voter choice in the neighboring counties. The influence of the neighboring regions is exactly what the spatial econometric analysis is capturing in contrast to the "non-spatial" studies that only recognize the impact of a broadly defined region per se. For instance, the fact that the Democratic Party earned 46.49% of the votes in the Gurye county of Jeollanam-do may be attributable both to the fact that this county is part of the Honam region and to the fact that the neighboring counties of Gokseong and Suncheon voted similarly, with the Democratic Party's voting shares being 49.4% and 41.78%, respectively. By taking separate accounts of the effects produced by the neighboring regions the spatial econometric analysis allows evaluation of the "pure" regional effect produced in voter choice.

## 4. Materials and Methods

### 4.1. Research Design

We start by providing basic summary statistics of our dataset and briefly discuss this in Section 4.2. In Sections 4.3 and 4.4 we provide a detailed description of Moran's $I$ and Getis-Ord $G_i^*$ that measure the extent of spatial autocorrelation and unexpected spatial spikes in the data, respectively. We provide ample evidence of statistically significant spatial dependence in our data in Section 5.1. Statistically significant spatial spikes are documented in Section 5.2. The strong presence of both spatial autocorrelation and spatial spikes warrants the application of spatial econometric analysis that we discuss in Section 4.5 where we also detail several spatial regression specifications. We discuss the results of statistical specification tests in Section 5.3. We present estimation results of the spatial econometric models in Section 5.4 and compare them to their OLS counterparts to argue that the OLS approach results in significant biases since it does not take into account the spatial spillover effects. Figure 4 below details the flow of our research.

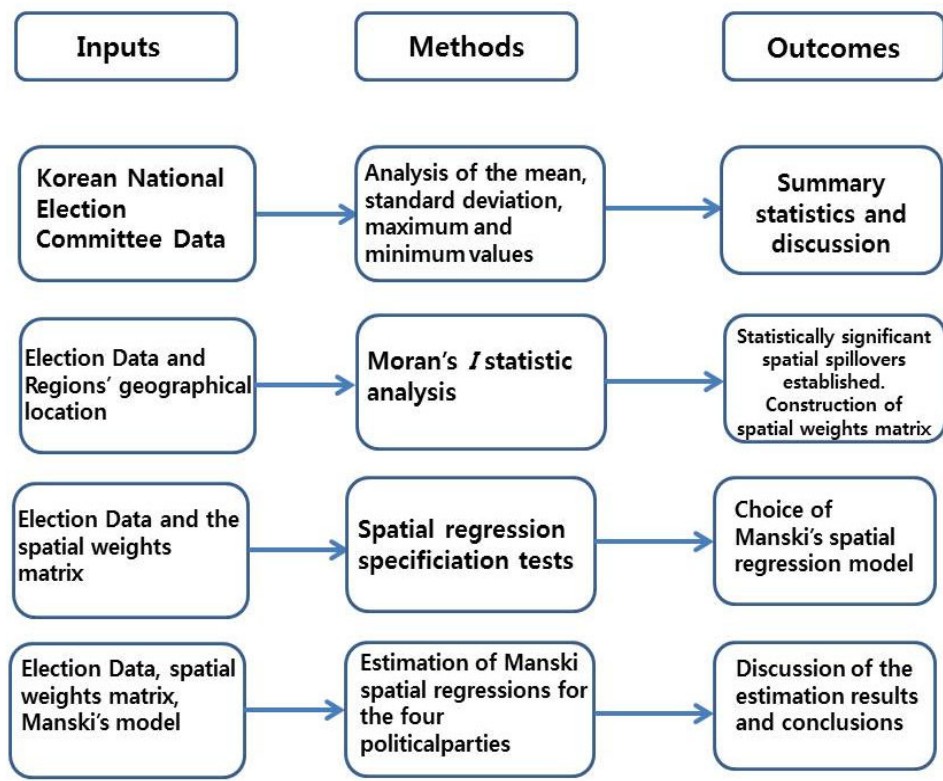

**Figure 4.** Research Flow.

*4.2. Data Sources and Summary Statistics*

The source of our dataset is Korea's National Election Commission that provides vote shares for 229 Korean regions. Most regions are reported as *gun* or *si* with the latter being more urbanized compared to the former. In case of the major metropolitan areas such as, e.g., Seoul or Busan, the regions are classified as autonomous *gu*, or urban districts. The socioeconomic indicators are collected from Korea's statistical yearbooks published by the National Statistical Office of Korea.

Table 1 provides summary statistics for our dataset.

**Table 1.** Summary Statistics.

|  | Average | Standard Deviation | Minimum | Maximum |
|---|---|---|---|---|
| | Voting Shares, % | | | |
| Saenuri Party | 33.50 | 17.13 | 2.63 | 70.83 |
| Democratic Party | 25.54 | 6.55 | 7.49 | 42.07 |
| People's Party | 26.74 | 11.15 | 9.91 | 59.56 |
| Justice Party | 7.23 | 2.01 | 2.74 | 13.57 |
| | Voting Behavior Determinants | | | |
| **Economic Characteristics** | | | | |
| **Regional GDP per capita, mil won** | 34.70 | 30.83 | 8.11 | 379.64 |
| **Unemployment rate** | 2.61 | 1.32 | 0.10 | 5.50 |
| **% employed in manufacturing** | 21.10 | 16.41 | 0.93 | 65.09 |
| **% employed in wholesale and retail trade** | 16.82 | 5.39 | 7.56 | 36.11 |
| **Budget balance, %** | 22.62 | 6.99 | 6.60 | 48.68 |
| **Labor force participation** | 54.74 | 5.30 | 39.68 | 73.03 |
| Socio-Demographic Characteristics | | | | |
| **Share of single-member households** | 36.41 | 9.06 | 18.25 | 64.71 |
| **Share of high-school graduates** | 71.17 | 10.25 | 34.83 | 94.76 |
| **Share of university graduates** | 40.99 | 16.22 | 17.65 | 86.16 |
| **Share of households with three or more children** | 3.39 | 0.81 | 1.60 | 6.71 |
| **Share of population older than 65** | 18.58 | 8.15 | 6.59 | 38.63 |

The winning Saenuri Party garnered an average of 33.50% of the votes while the two closest competitors, i.e., the Democratic and People's Party, earned an average of 25.54% and 26.74% of the votes, respectively. The Justice Party was able to collect a meager 7.23% of the votes on average making it a rather marginal competitor.

An average Korean region boasts a regional GDP per capita of about 35 million won, with the unemployment rate clustered around the 3% level. Regional GDP per capita in Honam reaches the level of 31.6 million won, which is not statistically different from the average value of 32.8 million won in Yeognam. This is especially interesting in light of the often-discussed topic of the alleged economic disparities between these two historically rival regions.

The share of single-member households in Korean regions is an alarming 36% on average, reflecting the recent tendency by young Koreans to defer marriage until reaching financial and career independence. For similar reasons, and in line with the well-recognized process of aging in Korean society, only about 3% of Korean households choose to have three or more children, contributing to the problem of population decline. Finally, the share of people older than 65 reaches an alarming average of 19%.

### 4.3. Moran's I Statistics to Test for the Presence of Spatial Spillover Effects in the Data

While Figure 3 suggests that political party support is regionally clustered, the statistical significance of these geographic patterns is not clear. Moran (1948) suggests two ways of verifying whether the nearby regions' characteristics are related in a statistically significant sense, i.e., whether they are spatially clustered. His approach can be summarized as follows.

Suppose we are analyzing $N$ regions indexed by $i = 1 \ldots N$. Denote $x_i$ to be a quantitative characteristic of region $i$ with the mean $\overline{x} = \frac{\sum_{i=1}^{N} x_i}{N}$ computed across all regions. In our context $x_i$ is the voting share in region $i$ for a particular political party. Denoting $z_i = x_i - \overline{x}$, consider the following statistic:

$$I = \frac{1}{\frac{1}{N}\sum_{i=1}^{N} z_i^2}\left[\sum_{j=1}^{N} w_{1j}z_1z_j + \sum_{j=1}^{N} w_{2j}z_2z_j + \ldots + \sum_{j=1}^{N} w_{Nj}z_Nz_j\right] \tag{1}$$

where $w_{ij}$ are elements of the spatial matrix $W^{N \times N}$ to be discussed shortly and are row-standardized, i.e., $\sum_{j=1}^{N} w_{ij} = 1 \ \forall i = 1 \ldots N$.

Each summand in the squared brackets in (1) is called local Moran's $I$ statistic, while the whole sum in (1) is referred to as (global) Moran's $I$. The latter is positive in the case that the neighboring regions vote similarly in the sense that a party gaining a high voting share in a particular region is likely to gain a similarly high voting share in neighboring regions in the country as a whole. Global Moran's $I$ statistic, however, captures the presence of spatial clustering in the complete geographical distribution of the voting shares without giving an indication as to the exact location of such voting clusters. It is exactly for the purpose of locating such spatial clusters that the local Moran's $I$ statistic is used.

The spatial weights $w_{ij}$, $i, j = 1 \ldots N$ assume nonzero values in the case that region $i$ is considered a neighbor of region $j$ in some sense with $w_{ii} \equiv 0$, i.e., no region is considered its own neighbor. LeSage and Pace (2009) review several ways in which spatial weights can be defined such as, e.g., queen contiguity whereby two regions are adjacent if they share a common border or corners in the polygon representation of regions, or rook contiguity where only sharing one or more borders matters. As our empirical results do not depend on the particular type of contiguity, in this study we report those based on queen contiguity.

The values of the local Moran's $I$ statistics are interpreted similarly to the values of global Moran's $I$, but in contrast they are computed for each region individually. Statistical significance for both global and local Moran's $I$ statistics is established by computing their

values for a large number of permutations which results in the pseudo-p values whose interpretation is the same as that for the conventional *p*-values. We discuss estimated values of Moran's *I* statistic in Korean regions in Section 5.1.

### 4.4. Getis-Ord Statistic to Identify Areas of High and Low Voting Shares

While Moran's *I* statistic helps identify those areas where voters in the adjacent regions vote similarly to their neighbors, it does not identify unexpected spatial spikes in either direction in the value of the voting shares relative to the sample average. A statistic developed by Getis and Ord (1992) allows one to identify such spikes along with the level of their statistical significance. When the voting shares in a particular location are significantly higher or lower compared to the expected (average) value, it results in a statistically significant z-score.

Technically, the Getis-Ord statistic is computed as follows:

$$G_i^* = \frac{\sum_{j=1}^{N} w_{ij} x_j}{\sum_{j=1}^{N} x_j} \tag{2}$$

The z-values for Getis-Ord statistics are computed, similarly to those of the Moran's *I* statistic, by calculating the values of $G_i^*$ in (2) for a large number of permutations. We discuss clustering maps suggested by the computation of $G_i^*$ in Section 5.2.

### 4.5. Spatial Models and Specification Tests

Our base empirical specification is as follows:

$$V_i = \alpha + \vec{\beta}' \vec{Z}_i + \varepsilon_i \tag{3}$$

where $V_i$ is the voting share gained by a particular political party in the general election of 2016 in region $i = 1 \dots 229$, $\vec{Z}_i$ is a vector of socioeconomic characteristics of region *i*, and $\varepsilon_i$ are independently and identically distributed random variables.

Luc Anselin (1988) demonstrated that in case spatial clustering is present in the data, a standard (OLS) linear regression approach to estimating the (base) parameters may result in biased and inconsistent estimates due, e.g., to the omitted variable bias in case of ignoring spatial lags.

In principle, there are three major ways in which spatial correlation may be present in the data. First, values of the dependent variable in a certain region, $V_i$ in our context, may depend on its own values in the surrounding regions, a phenomenon commonly referred to as spatial autocorrelation. One illustration of spatial autocorrelation is provided by Figure 3 in Section 3.4. Second, the region-specific errors in (2) may be a function of their values in the neighboring regions, in which case the errors in (2) are said to be spatially correlated. Modifying (2) to account for this issue results in a spatial error model (SEM). Third, spatial correlation may also be present in the independent variables comprising $\vec{Z}$. The abovementioned three types of spatial correlation are not mutually exclusive so that, e.g., a spatially correlated dependent variable may coexist with the spatially correlated independent variables, or with the spatially correlated errors, or with both.

Elhorst (2012) formulates an umbrella model that takes into account the various ways in which data may be spatially correlated. His formulation is as follows:

$$\begin{cases} \vec{y} = \rho W \vec{y} + Z\vec{\beta} + WZ\vec{\theta} + \vec{u} \\ \vec{u} = \lambda W \vec{u} + \vec{\varepsilon} \end{cases} \tag{4}$$

where $\vec{\varepsilon}$ is independently and identically distributed.

Spatial lags of the dependent variable $y$ and the independent variables $\overrightarrow{Z}$ are given by $W\overrightarrow{y}$ and $WZ$, respectively. $W\overrightarrow{u}$ is a vector of weighted averages of errors in the neighboring regions. The specification (3) turns into a classical linear regression model if $\rho = \lambda = 0$ and $\overrightarrow{\theta} = 0$. Table 2 details the six models comprised by (3) that we consider in this study.

**Table 2.** Taxonomy of the Spatial Models' Specifications.

| Parameter Values in (Elhorst) | | | Model |
|---|---|---|---|
| $\rho$ | $\overrightarrow{\theta}$ | $\lambda$ | |
| $\neq 0$ | $= 0$ | $= 0$ | Spatial autoregression (SAR) |
| $\neq 0$ | $\neq 0$ | $= 0$ | Spatial Durbin (SDM) |
| $= 0$ | $= 0$ | $\neq 0$ | Spatial error (SEM) |
| $= 0$ | $\neq 0$ | $\neq 0$ | Spatial Durbin error (SDEM) |
| $\neq 0$ | $= 0$ | $\neq 0$ | Kelejian and Prucha (KP) |
| $\neq 0$ | $\neq 0$ | $\neq 0$ | Manski (M) |

We do not consider a model where spatial lag is only present in the independent variable, i.e., the one characterized by $\rho = \lambda = 0$ and $\overrightarrow{\theta} \neq 0$, as to our knowledge it has not been a popular choice in the spatial econometric analysis. The (KP) model is developed in (Kelejian and Prucha 1998), while the most comprehensive Manski model is analyzed in (Manski 1993). We present the results of statistical tests on the best spatial specification in Section 5.3.

## 5. Results

### 5.1. Statistical Tests for the Presence of Spatial Effects

As discussed in Section 4.3, the presence of spatial spillover effects in the data is indicated by the statistically significant values of the Moran's *I* statistic. The value of global Moran's *I* statistic is estimated to be equal to 0.85 for the spatial distribution of voting shares for the Saenuri Party, 0.68 in case of the Democratic Party, 0.87 in case of the People's Party, and 0.55 in case of the Justice Party. In all four cases the value of the pseudo-p is evaluated at or below 0.001 based on 999 permutations, implying strong spatial correlation in the geographical distribution of voting shares for all of the four competitors in Korea's 2016 general election.

Figure 5 below presents cluster maps that illustrate the geographical distribution of the statistically significant local Moran's *I* statistics for the three major competing parties. In the red-colored regions the voting share of a particular political party is relatively high, and is highly correlated with that in the neighboring regions in the statistical sense. In contrast, the blue-colored regions are characterized by a low level of support for the same party as well as by a low statistical significance of the spatial correlation with the voting shares in the neighboring regions.

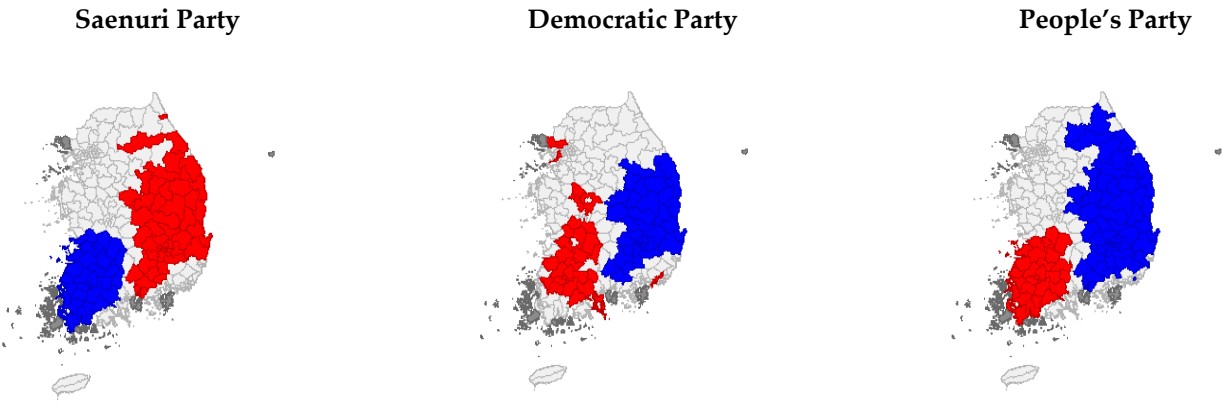

**Figure 5.** Moran's *I* statistic in Korean regions.

The three cluster maps above suggest that statistically significant clustering of Korean voter behavior is mostly a feature of the two perennially antagonistic regions, namely, the South-Western Honam, and the South-Eastern Yeongnam. In case of the Saenuri and the People's parties, spatial clustering is also a feature of a certain number of regions in Korea's North-Eastern Gangwon-do province, which is not the case for the Democratic Party. In the rest of the country, including Seoul metropolitan area, voting behavior does not appear to be spatially clustered in a statistically significant sense.

### 5.2. Identification of Spatial Spikes in the Spatial Distribution of Voting Shares

As discussed in Section 4.4, the Getis-Ord $G_i^*$ statistic allows one to identify hot spots, which in our context means areas where the voting shares in a particular region and its adjacent regions are significantly higher in the statistical sense compared to the country's average. Cold spots are defined in a mirror fashion. In Figure 6 we present hot and cold spot cluster maps computed on the basis of the Getis-Ord $G_i^*$ statistic.

| **Saenuri Party** | **Democratic Party** | **People's Party** |
|---|---|---|

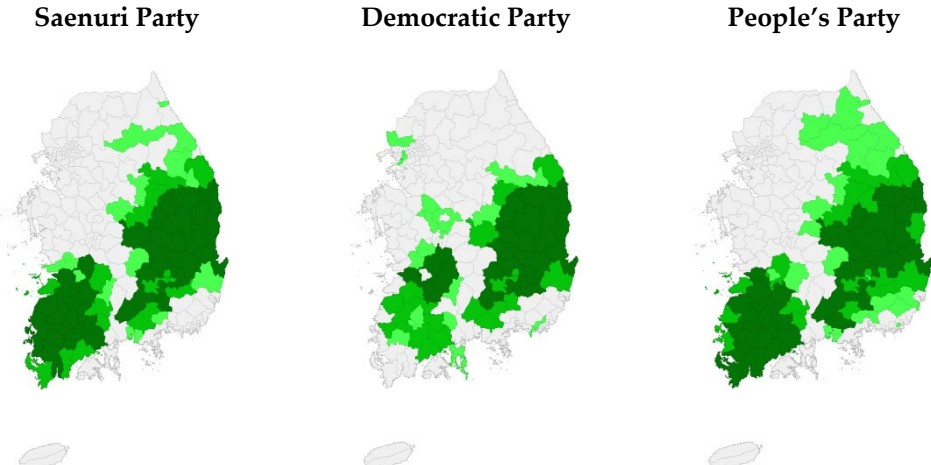

**Figure 6.** Getis-Ord $G_i^*$ statistic in Korean regions.

The darker areas in Figure 6 correspond to those regions where the difference between voting shares for a particular political party in a certain area is greater in a statistically significant sense compared to the expected value for the country as a whole. Two observations are worthwhile. First, we observe a high concentration of the dark areas in the Honam and Yeongnam regions for all three political parties. Second, the areas where the values of the Getis-Ord $G_i^*$ statistic are significantly different from the mean in the statistical sense virtually coincide with the red- and blue-colored areas in Figure 5. It is important to notice that the voting shares for all parties in Honam and Yeongnam are significantly higher or lower than the country mean whether or not the spatial correlation effects are present. For instance, the Honam voters do not support the Saenuri party, which is why Honam is a dark area in Figure 6. At the same time, Honam is a blue area in Figure 5, which means spatial correlation effects in Honam are weak according to the Moran's *I* statistic. In other words, voters' proclivity to overwhelmingly support or dislike a political party is not necessarily associated with the strong spatial autocorrelation effects, which is another reason why it is important to take into account the influence of the neighboring regions in our analysis.

### 5.3. Results of the Spatial Specification Tests

Our objective in this section is twofold. First, we argue that spatial effects are indeed present in the data so that statistical inference using one of the spatial regression specifications summarized in Table 2 is justified. Second, we identify the spatial regression specification which most closely fits our data. In both cases we rely on the likelihood ratio

tests described in (Bivand and Piras 2015). We conduct these tests in the R environment using the "spdep" package of (Bivand 2016).

In Table 3 we list the log-likelihood (LL) and Akaike information criterion (AIC) values for each spatial regression specification with the voting share of the Saenuri Party as a dependent variable. We also report the LL and AIC values for the benchmark OLS model. The likelihood ratio test statistics along with their significance levels are reported for each pairwise comparison between the two spatial regression specifications along with the name of the statistically preferred model. Log-likelihood ratio tests are not applicable in case the two models are not nested. In this case our choice of the preferred model is guided by the values of LL and AIC.

**Table 3.** Spatial Model Specification Tests for the Saenuri Party Voting Shares.

| | | | Log-Likelihood Ratio Test | | | | | |
|---|---|---|---|---|---|---|---|---|
| | **LL** | **AIC** | **SEM** | **SAR** | **SDM** | **KP** | **SDEM** | **Manski** |
| **Benchmark** | −758 | 1544 | SEM (201.3) *** | SAR (177.3) *** | SDM (225.1) *** | KP (203.8) *** | SDEM (212.0) *** | Manski (231.3) *** |
| **SEM** | −657 | 1345 | NA | NA | SDM (23.8) ** | KP (2.5) | SDEM (10.7) | Manski (30.0) *** |
| **SAR** | −669 | 1369 | NA | NA | SDM (47.8) *** | KP (26.5) *** | NA | Manski (54.0) *** |
| **SDM** | −645 | 1345 | SDM (23.8) ** | SDM (47.8) *** | NA | SDM (21.3) *** | NA | Manski (6.2) ** |
| **KP** | −652 | 1344 | KP (0.6) | KP (26.5) *** | SDM (21.3) *** | NA | NA | Manski (27.5) *** |
| **SDEM** | −649 | 1358 | SDEM (7.4) | NA | NA | NA | NA | Manski (19.3) *** |
| **Manski** | −642 | 1341 | Manski (27.8) ** | Manski (54.0) *** | Manski (6.2) ** | Manski (27.5) *** | Manski (19.3) *** | NA |

Note: Values of the absolute value of the log-likelihood ratio test statistic are reported in parentheses under the name of the preferred specification in a pairwise comparison. ***, **, and * denote 1%, 5%, and 10% significance levels, respectively. Likelihood ratio test is only applicable to the nested models so that the value of the LR statistic in choosing between, e.g., SAR and SEM are incomputable.

The first row of Table 3 implies that econometric specifications in (Elhorst 2012) with spatial effects outperform the benchmark OLS specification. The latter also has the lowest value of the log-likelihood and the highest value of the Akaike information criterion statistic associated with it. We conclude that spatial effects are strongly present in our sample, thus justifying econometric estimation with spatial effects.

Among the six spatial specifications summarized in Table 3, Manski's portmanteau specification is characterized by the highest value of the log-likelihood function, and by the lowest value of the Akaike information criterion statistic. In addition, as evidenced by the last column of Table 3, Manski specification outperforms its five competitors in terms of the likelihood ratio test at least at a 5% statistical significance level.

While the data in Table 3 refer to the analysis of the various specifications in (3) with the voting share of the Saenuri Party as a dependent variable, we also get the same qualitative results in case of the voting shares of the remaining three political parties. We thus find it justified to continue our analysis by estimating the comprehensive Manski specification in all cases. The specification test results for the three remaining political parties are available from the authors immediately upon request.

### 5.4. Estimation Results of the Spatial Models

In Table 4 below we present our estimation results of the most comprehensive Manski specification (see (3) and Table 2) for the four political parties that competed in the 2016 Korean general election.

**Table 4.** Determinants of Koreans' Voting Choice in the General Elections of 2016.

| | Saenuri Party | Democratic Party | People's Party | Justice Party |
|---|---|---|---|---|
| **Preferred Model** | **Manski** | **Manski** | **Manski** | **Manski** |
| **Dependent Variable** | **Share of Votes in a County** | | | |
| **Constant** | −2.963 (−0.366) | 2.231 (0.116) | 13.230 (2.430) ** | 3.463 (1.240) |
| **RGDPC** | 0.212 (0.349) | −0.667 (−1.135) | 0.373 (0.797) | 0.113 (0.537) |
| **U** | 0.258 (0.445) | −0.371 (−1.238) | 0.061 (0.236) | 0.052 (0.515) |
| **MFSH** | −3.514 (−1.275) | 2.540 (1.013) | 1.340 (0.630) | −0.434 (−0.522) |
| **SERVSH** | −7.006 (−0.823) | 3.196 (0.429) | 8.899 (1.356) | 0.968 (0.378) |
| **LPART** | **−19.073 (−2.177) **** | **32.121 (3.864) ***** | −7.554 (−1.105) | −0.308 (−0.117) |
| **OLDSH** | **63.232 (7.646) ***** | **−39.874 (−5.440) ***** | **−24.398 (−3.851) ***** | **−13.046 (−5.232) ***** |
| **BUDBAL** | **12.181 (2.908) ***** | 1.597 (0.404) | **−8.127 (−2.501) **** | **−2.803 (−2.228) **** |
| **SINGSH** | −2.810 (−0.541) | −3.628 (−0.750) | 5.304 (1.319) | −2.055 (−1.317) |
| **GOJOLSH** | **6.031 (1.711) *** | −1.151 (−0.384) | −0.394 (−0.145) | **−4.488 (−4.228) ***** |
| **DASANSH** | −87.870 (−1.229) | 19.007 (0.327) | 6.626 (0.120) | −14.803 (−0.691) |
| **YEONGNAM** | **11.477 (7.954) ***** | **−6.935 (−6.265) ***** | **−4.413 (−4.027) ***** | −0.029 (−0.067) |
| **HONAM** | **−21.745 (−11.128) ***** | **7.667 (5.039) ***** | **11.383 (7.796) ***** | 0.865 (1.462) |
| $\lambda$ | −0.399 *** | 0.845 *** | −0.584 *** | −0.310 |
| $\rho$ | 0.883 *** | −0.323 ** | 0.888 ** | 0.623 *** |

Note: z-values are in parentheses. ***, **, and * stand for the 1%, 5%, and 10% significance levels, respectively. Estimates for the spatial lags of independent variables are omitted. The voting choice determinants are explained and summarized in Section 2.1.

In case of all four political parties our estimation results imply that the dependent variable, i.e., these parties voting shares, are spatially correlated at a five percent statistical significance level. The errors *u* in model (3) are also estimated to be spatially correlated in all cases except for the Justice Party, a marginal competitor to the other three, thus confirming the need to employ spatial econometric analysis as opposed to the non-spatial analysis tools. In what follows we summarize our estimation results whose detailed discussion will be provided in Section 6.

The one determinant that is consistently estimated to be statistically significant in case of all four political parties is the share of population older than 65 years. Older voters appear to be in favor of the winning Saenuri Party but not of the Democratic, People's, and Justice parties.

We find it remarkable that economic characteristics of the Korean regions do not appear to be producing an important influence on the voters' choice as the coefficients on the regional GDP per capita, unemployment rate, or the share of manufacturing employment are not estimated to be statistically significant in any one of the four estimated models.

We only find two economic characteristics to be of some importance to the voter choice in our analysis, namely, the local governments' budget balance, and the regional labor force participation. The coefficient for the regional budget balance is estimated to be statistically significant in all cases except for the Democratic Party. The sign of the effect, however, is not uniform across the political spectrum. Thus, the coefficient for the budget balance is estimated to be positive for the winning Saenuri Party, while it is negative in case of

the Justice and People's parties. The coefficient for the labor force participation is only estimated to be statistically significant in case of the Saenuri (positive), and the Democratic Party (negative).

The Yeongnam and Honam dummies are estimated to be statistically significant for the Saenuri, People's, and Democratic parties, but not for the Justice Party. Our findings are consistent with the results presented by (Lee and Repkine 2020) in the context of the Korean Presidential election of 2017, where the voters' region of origin and their age were also found to be of paramount importance in determining voters' choice. This similarity in results is all the more important, not only because our study is addressing a different type of election, but also in that we are using the data collected at a regional level, while the study of (Lee and Repkine 2020) is based on the voters' survey.

As discussed in Section 1.2, statistical significance of the Honam and Yeongnam dummies which is still there even after the spatial spillover effects have been taken into account is indicative of the presence of "pure" regional effects in the Korean voting behavior. In other words, Korean voter choice in Honam and Yeongnam is driven in large part by the regional sentiment alone that is unrelated either to the socio-economic characteristics of the voters' regions or the voter choice in the neighboring regions. Since the influence of the latter can only be captured by the spatial econometric analysis, the non-spatial estimation frameworks such as, e.g., OLS are likely to confound the regional sentiment effect with that of the neighboring regions, thus likely overstating the importance of the regional sentiment effect per se. To illustrate the value of accounting for the influence of the neighboring regions with the spatial econometric analysis, we report the OLS estimates of the models in Table 5 below:

**Table 5.** The OLS Modeling of Koreans' Voting Choice, the General Elections of 2016.

| | Saenuri Party | Democratic Party | People's Party | Justice Party |
|---|---|---|---|---|
| **Dependent Variable** | Share of Votes in a County | | | |
| **Constant** | 46.729 (4.49) *** | 18.260 (2.87) *** | 18.786 (2.40) ** | 12.692 (6.33) *** |
| **RGDPC** | −0.003 (−0.17) | −0.005 (−0.46) | 0.004 (0.30) | 0.004 (1.24) |
| **U** | −0.761 (−1.42) | −0.394 (−1.20) | **0.972** **(2.41) **** | 0.146 (1.42) |
| **MFSH** | −0.248 (−0.05) | 3.169 (1.11) | −0.957 (−0.27) | −1.098 (−1.22) |
| **SERVSH** | −8.560 (−0.62) | 10.448 (1.23) | 9.567 (0.91) | −0.028 (−0.01) |
| **LPART** | −42.777 (−3.02) *** | 22.719 (2.62) *** | 13.596 (1.27) | 3.760 (1.38) |
| **OLDSH** | 73.310 (5.76) *** | −31.387 (−4.02) *** | −32.561 (−3.39) *** | −18.000 (−7.34) *** |
| **BUDBAL** | 21.723 (3.09) *** | **−8.724** **(−2.02) **** | −10.948 (−2.06) ** | −3.296 (−2.43) ** |
| **SINGSH** | 1.709 (0.20) | −5.442 (−1.05) | 6.192 (0.97) | **−3.378** **(−2.07) **** |
| **GOJOLSH** | 12.024 (2.35) ** | −2.121 (−0.68) | **−7.908** **(−2.05) **** | −3.523 (−3.58) *** |
| **DASANSH** | **−207.831** **(−2.73) **** | **93.002** **(2.00) **** | **116.374** **(2.03) **** | **−41.737** **(−2.85) **** |
| **YEONGNAM** | 11.274 (8.12) *** | −6.725 (−7.91) *** | −5.010 (−4.79) *** | −0.164 (−0.62) |
| **HONAM** | −38.070 (−23.09) *** | 8.348 (8.27) *** | 25.665 (20.65) *** | **1.705** **(5.37) **** |
| **Adjusted R²** | 84.07% | 59.08% | 78.62% | 56.91% |

Note: t-values are in parentheses. ***, **, and * stand for the 1%, 5%, and 10% significance levels, respectively. The voting choice determinants are explained and summarized in Section 2.1. Highlighted in bold are the coefficients that are estimated to be statistically significant within the OLS, but not the spatial econometric framework.

Within the OLS estimation framework the coefficients of some socio-economic variables are estimated to be statistically significant as opposed to the spatial econometric estimates presented in Table 4. The coefficient for the share of households with three or more children, for instance, is estimated to be statistically significant in all of the four cases with the OLS in contrast to the spatial econometric estimation. In case of the People's Party estimates, the unemployment rate and the share of voters who graduated from high school is estimated with a statistically significant coefficient by the OLS, but not by the spatial econometric approach. The OLS similarly attaches statistical significance to the coefficient on the budget balance variable. Even when the OLS agrees with the spatial econometric models on the statistical significance of the estimated coefficients, the difference in magnitude can be significant. Thus, the OLS coefficient of labor force participation in case of the Saenuri party is estimated at the level of $-42.8$, while the Manski model suggests the value of $-19.1$, a difference of more than 100%. In case of the share of older voters, the OLS estimate of the coefficient for this variable in case of the Saenuri party is positive, while it is negative in the spatial econometric framework.

Finally, the OLS estimates of the coefficients for the regional dummy variables are in general larger in magnitude compared to their spatial econometric counterparts, especially in case of the Honam. Thus, in case of the People's party the Honam dummy coefficient is estimated at the level of 25.7 by the OLS, while the estimate is 11.4, or less than half that in case of the spatial econometric estimation. The same is true for the Saenuri party. Importantly, in case of the Justice party the Honam dummy coefficient is estimated to be statistically significant by the OLS while it is not in case of the spatial Manski model. We take these results as evidence of the fact that the spatial econometric approach is beneficial as opposed to the non-spatial estimation frameworks in at least two respects. First, by accounting for the influence of the neighboring regions it removes the effect of their influence from the other explanatory variables, including the regional dummies. Second, as a result of such removal the regional dummy coefficients can be interpreted as representing the "pure" regional effect that in our context refers to the psychological, or sentimental, attachment of voters to their region. In contrast, the OLS-like, or a non-spatial approach in general, is confounding these two effects, namely, the influence of the neighboring regions, and the influence of the region per se.

## 6. Discussion

Our empirical results imply that voters' age is a major characteristic affecting voter choice with older voters consistently favoring the more conservative Saenuri Party and younger voters tending to support its more progressive competitors. As noticed by (Kang 2003), the ideological cleavages in Korea are shaped much more by the inter-generational conflict as opposed to conflicts between economic classes, compared to the Western economies. Our results also confirm the importance of age in shaping Korean voter choice indirectly, as we find no statistical significance in the coefficients of such important economic characteristics as the rate of unemployment and regional GDP per capita. Our finding that older voters prefer a more conservative Saenuri Party accords well with the observation by (Jeong 2020) who notices that older voters in Korea tend to favor the more conservative parties compared to younger ones. What is remarkable is that we estimate the age effect to remain strong even after we have controlled for the several types of interregional spillover effects.

The well-known reward-punishment model of voting behavior predicts that those political parties that are perceived to be improving voters' economic conditions will get a larger share of the votes, see, e.g., (Tillman and Park 2009). This theory has found support in the international context. Thus, Kim et al. (2003) find a significant effect of the unemployment rate on voter choice in the US counties during the US presidential elections between 1988 and 2000. Caleiro and Guerreiro (2005) come to a similar conclusion when conducting a spatial econometric analysis of the Portuguese legislative elections of 2002. In the Korean context, however, the socio-economic characteristics appear to produce less

pronounced an influence on voter choice compared to the Western democracies. Thus, Lee and Repkine (2020) conducted a spatial econometric analysis of the Korean presidential elections of 2017 and found that among the various regional-level economic indicators it is only the share of employment in manufacturing that produced a statistically significant effect on voting behavior. The empirical results of the present study are consistent with this finding, as none of the coefficients on the unemployment rate, regional GDP per capita, or the employment shares in manufacturing and the service sector are estimated to be statistically significant. To reiterate, these findings are in stark contrast with what is known as the phenomenon of "economic voting" prevalent in the Western democracies, see, e.g., (Duch 2008) and (Lewis-Beck and Stegmaier 2008).

One explanation why socio-economic indicators appear to play such a small role in shaping Korean voters' choice may be that Korea is a young democracy with a relatively short history of party politics conditioned by a legacy of the decades-long standoff between the ruling and the opposition party prior to the event of democratization of 1987. In other words, starting from the late 1980s, the Korean political parties were characterized by a relatively weak support base that, among other factors, had to do with the fact that different political parties in Korea were not characterized by much variation in their economic policies, see (Park 2019). In this context, our finding of the weak association between regional economic characteristics and voter choice in Korea should be interpreted as resulting from a lack of partisan variation in the country, especially in regard to these parties' economic agendas, rather than from the lack of Korean voters' sensitivity to economic well-being.

We believe our empirical results imply that a positive budget balance of a local government in Korea acts as a signaling device on whether the incumbent party is pursuing conservative or progressive policies. Indeed, studies such as, e.g., (Choi and Park 2014), (Burch and Wood 1990) and (Boyne 1991) argue that, since more progressive political parties put more emphasis on welfare spending, local government run by a progressive political party member will be likely characterized by a low or a negative budget balance. Conversely, a conservative political party would emphasize sound fiscal policies while avoiding excessive spending on welfare, resulting in higher values for the budget balance, see (Jee and Kim 2003). As a result, the more conservative parties would obtain more votes when running a balanced or a positive budget balance, while the opposite is true for the more liberal, or progressive, political parties. This line of reasoning fits rather well with our empirical finding of a positive coefficient on the budget balance in case of the conservative Saenuri Party, and the negative coefficients on the same variable for the more liberal, or progressive, Justice and People's parties. To summarize, frugal spending of public funds appears to serve as a signal in the Korean context of a party conducting more conservative policies, thus luring conservative voters, while the opposite is true for the more liberal political parties and their support base.

The Yeongnam and Honam dummies are estimated to be statistically significant for the Saenuri, People's, and Democratic parties, but not for the Justice Party. The Honam residents, expectedly, do not favor the conservative Saenuri Party with the opposite being true of the Yeongnam residents. This finding is all the more interesting since the Manski model we estimate for all four political parties takes into account three possible types of spatial correlation in the data, which implies that statistically significant regional dummies of Honam and Yeongnam capture the "pure" regional effect, i.e., the effect produced by the region per se. In other words, a large part of the reason why, for instance, a voter residing in a Honam county votes for the Democratic Party is because of a regional sentiment that can be the result of, e.g., political mobilization in the absence of alternative social cleavages, as discussed in Section 1.2. The presence of such regional sentiments and the strength of their influence on Korean voter choice is potentially threatening to democratic rule, as competition between political and socio-economic agendas between political parties is facing the risk of being replaced by a competition between regional attachments.

As in all spatial econometric analyses, ours is based on a particular partition of Korea into a set of government-defined voting constituencies, which raises the question of sensitivity of our results to changes in such partition. One interesting exercise would be to conduct our analysis at a more disaggregated level of, e.g., eight Korean provinces or even at a level of larger regional swathes that would include the broadly defined regions of Honam and Yeongnam. To our knowledge, there is no established way of assessing the extent to which the results of a spatial econometric analysis are sensitive to the definition of regions, so that developing such a sensitivity framework is a promising future research avenue.

We believe Korean election outcomes provide an excellent ground for the spatial econometric analysis conducted in this study based on historical, factual, and statistical grounds. Historically, as reviewed in Section 2.2, Korea has a centuries-old legacy of conflict between the regions of Honam and Yeongnam. Factually, this conflict did not disappear even as Korea underwent impressive democratic changes since 1987, with regionalism in voting behavior only becoming stronger. Statistically, we demonstrate that significant spatial spillover effects are indeed present in our data (Section 5.1) and that these effects significantly alter the model estimation results compared to the OLS-type, non-spatial, econometric techniques, as discussed in Section 5.4. Compared to other ethnically homogeneous countries such as, e.g., Japan, Iceland or Portugal, Korea appears to be unique in that it shows a strong regional conflict, thus warranting the application of spatial econometric techniques.

However, spatial econometric analysis has proven to be useful also in those democracies where the primary electoral cleavage is not based on regional divisions. Thus, Caleiro and Guerreiro (2005) apply a spatial approach to the analysis of the Portuguese legislative elections in 2002 and find evidence of significant spatial spillover effects even in the absence of a regional division problem of the Honam–Yeongnam type. In certain cases spatial clustering of the voter camps may occur along ideological lines with the same type of ideology not necessarily represented by a single contiguous region, as is well evidenced by the existence of the so-called red and blue states in the US, see e.g., (Miller and Conover 2015).

In the context also of the century-long data on the US presidential elections, Braha and De Aguiar (2017) estimate a model of social contagion and conclude that the voters' tendency to imitate each other is spatially clustered, which again does not automatically imply a one-to-one correspondence between any one contiguous region and a certain ideology. While our study demonstrates the value of the spatial econometric approach in Korea where regionalism is a well-known phenomenon, the spatial clustering of voting behavior is a more general tendency that should be analyzed also in economies where regionalism is not a salient issue.

## 7. Conclusions

We applied spatial econometric analysis to a unique dataset on the outcomes of the 2016 general elections in Korea at a highly disaggregate level of 229 provinces comprising the country. Statistical tests have confirmed that spatial spillover effects are strongly present in our sample thus justifying the choice of those empirical specifications that take into account spatial autocorrelation in the data.

Our empirical findings align well with the existing evidence on Korean voting behavior, in particular regarding the influence produced by the voters' region of origin, and their age. Thus, the winning Saenuri Party is estimated to be mostly favored by the older, and hence conservative, voters residing in Korea's South-Eastern region of Yeongnam. In contrast, the younger voters residing in the South-Western region of Honam appear to prefer the People's and Democratic parties which have collectively garnered about half of the votes in an average Korean region. The influence of geographical region does not appear to be present in the case of the Justice Party which can be classified as a rather marginal competitor in the general elections of 2016 since it only gained an average of 7% of the

regional votes. Older voters are also more likely to prefer the conservative Saenuri Party, while in contrast younger voters support of the Justice, Democratic, and People's parties.

Our empirical model controls for three types of spatial autocorrelation, namely, spatial dependence in both the dependent variable and the determinants of the voter choice, and in the model's error terms. Yet, the regional dummies for Honam and Yeongnam, Korea's historically antagonistic regions, are estimated to be statistically significant in case of the three major political parties, although not in case of the Justice Party. This finding implies that a voter's decision making is influenced both by the voter choice made in the adjacent smaller regions, and by the fact of being located in a relatively large region such as, e.g., Honam itself. In other words, even after controlling for the three types of interregional spatial dependence in the data, we still identify the "pure" regional effect given by the statistically significant Yeongnam and Honam dummies, thus adding to the evidence on the strength of political regionalism in Korea.

Differently from the similar studies conducted for the Western democracies, economic characteristics such as the regional income per capita or the rate of unemployment do not appear produce a statistically significant effect on Korean voters' choice. We take it as evidence of a lack of variation of the Korean political parties' economic agendas, which is not surprising given the relatively short history of Korean political parties. It is all the more interesting then to find a statistically significant effect of the budget balance variable that, as we have argued in the previous section, appears to be acting as a signaling device distinguishing between a conservative and a liberal political agenda. More research is needed in this area that would be based on the analysis of the welfare policies such as, e.g., income redistribution in order to test whether conservative parties are indeed spending less on such programs compared to their more liberal counteparts.

We believe our results are interesting in light of the growing concerns regarding the rapidly aging Korean society (see recent discussion in (Lee 2021a, 2021b)). Our finding of older voters leaning towards the conservative edge of the political spectrum suggests that the "silver democracy" now actively discussed in the Korean media is increasingly assuming more conservative traits. Given the tendency of the Honam residents to vote for the more liberal political parties, future Korean society is likely to become polarized along both the geographical, and the age dimension. Since, in contrast to studies such as (Lewis-Beck and Stegmaier 2000) who find economic characteristics produce an important influence on voter choice, these characteristics do not appear to be significant in the Korean context, the scope of government policies aimed at taming Korean society's regional polarization definitely deserves a study of its own.

**Author Contributions:** Conceptualization, A.R. and H.-C.L.; methodology, A.R.; formal analysis, A.R.; resources, H.-C.L.; data curation, H.-C.L.; writing—original draft preparation, A.R.; writing—review and editing, A.R. and H.-C.L.; visualization, H.-C.L. and A.R. All authors have read and agreed to the published version of the manuscript.

**Funding:** This research received no external funding.

**Informed Consent Statement:** Informed consent was obtained from all subjects involved in this study.

**Data Availability Statement:** The dataset used in this study is available from the authors.

**Conflicts of Interest:** The authors declare no conflict of interest.

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
