# Peer review of "A Spatial Analysis of the Voting Patterns in the South Korean General Elections of 2016"

_socsci, doi:10.3390/socsci11090389_

Round 1

Reviewer 1 Report

In this paper the authors apply spatial econometric analysis to the outcomes of the 2016 general elections in Korea to understand the spatial and socioeconomic factors that influence voters’ behavior. The most important finding in this paper is that voters’ choice is influenced by the voters’ choice made in adjacent regions, thus contributing to the understanding of political behavior in its spatial context. I would like to commend the authors for their excellent work in applying a comprehensive spatial analysis to the understanding of elections and voting behavior. Below, I would like to suggest several ways by which the paper can be improved.

Comment 1. The authors might benefit from the insights provided in a recent paper that analyzed a century of U.S. presidential elections (see reference below). More specifically, the paper below developed a model of social influence (i.e., the tendency of voters to imitate others), and applied it to a large-scale voting data of U.S. presidential elections. By applying spatial statistical data analysis methods, the paper provides comprehensive evidence that the tendency of voters to imitate others is spatially clustered, indicating the growing role of social influence, contagion, and ‘herd-following’ in shaping large-scale collective political behavior. This immediately implies that voting behavior in contiguous regions is autocorrelated. It would be useful if the authors can address these findings in the current paper as they provide the theoretical underpinning of Tobler’s law in general and voting behavior in particular.

Comment 2. While the Moran’s I statistic indicates that the spatial distribution of high and/or low values is more spatially clustered than would be expected if underlying processes were random, it does not identify unexpected spatial spikes of high or low social influence values (see reference below). It would be useful if the authors can extend their analysis by including tests of spatial clustering using the Getis-Ord General G* statistic (see reference below).

Reference

Braha, D., & De Aguiar, M. A. (2017). Voting contagion: Modeling and analysis of a century of US presidential elections. PloS one, 12(5), e0177970.

Author Response

We thank Reviewer 1 for his or her very useful comments.

Please find our detailed response in the file attached.

Reviewer 2 Report

This paper demonstrates the value of spatial analysis for accurately estimating what influences South Korean elections. Once spatial clustering is accounted for, the authors show that standard economic factors known to pivotally shape Western elections held little water in South Korea’s 2016 general election. Instead, region and generation emerge as the key factors.

I very much liked this paper and saw its contributions by the end. But the paper starts off weak. The authors need to better ground the substantive importance of applying spatial analysis to elections before jumping straight into Tober’s law and the statistical principles behind spatial econometrics. Spell out for us why we should pay attention to spatial analysis in this particular electoral context before getting into the nitty gritty. I suggest opening with a broader discussion about the shifting nature of electoral politics in South Korea (and perhaps neighboring democracies experiencing similar demographic change, such as Japan or Taiwan). Then explain why accurately understanding the pivotal determinants of electoral outcomes in these cases matters for things like regional stability, democratic rule, and so on. Now we are paying attention.

THEN begin to explain how spatial analysis can help get to more accurate estimates. This is certainly not the first paper to apply spatial econometrics to electoral results, so the authors need to 1) better contextualize their study within prior election studies, especially in American politics where the movement started, and 2) clarify how their approach makes a unique empirical or substantive contribution to South Korean electoral studies. Key citations here are missing: Poole and Rosenthal’s seminal 1984 AJPS piece and a large scholarship in Electoral Studies on the topic (Dow 2001, etc).

In looking through this literature, I cannot help but notice the near parallel setup, methods, and findings to an existing spatial study of Korean elections: Lee and Repkine’s “Changes in and Continuity of Regionalism in Korea: A Spatial Analysis of the 2017 Presidential Election Outcomes” in Asian Survey, 2020. The authors cite the work in passing, but do not engage with the meat of that study, which is its application of spatial analysis. If this study is meant to be an extension of that former piece (2017 presidential in Lee and Repkine vs. 2016 general in this study), then it is unclear that it is. There is real value to publishing replications and extensions of prior work, but those efforts need to be clearly framed as such. I leave this to the adjudication of the editor.

Some non-statistically inclined readers may find the discussion about the effect of regionalism vs. the “pure” regional effect post-spatial clustering to be confusing. This discussion appears on p. 9-10, but it is embedded in the middle of a paragraph and remains technical in nature. It needs to be front and center in the analysis and also theoretically deeper. The legacy of regional voting in South Korea is the result of many distinct but intertwined factors: the hometown effect of pivotal leaders, divergent economic investments from the government, competition for status, and socialization. Which of these mechanisms remains after spatial clustering? What does the “pure” regional effect capture and why does it matter? In general, the paper needs to do a better job linking the statistics with electoral and political substance. This connection gets better by the end of the paper, but it comes too late.

One way to empirically illustrate the value of spatial analysis would be to run the same model specifications as in Table 4 but in OLS with region dummies. Do the biased OLS estimates yield very different predictions about what factors mattered to the 2016 election? If, for instance, the OLS estimated that economic factors mattered a lot (in direct contrast to what the authors find with spatial clustering), that would clearly demonstrate the value of spatial econometrics.

Finally, the study warrants a nuanced discussion on scope conditions. To what extent are South Korean elections a particularly good candidate for spatial analysis? Would spatial econometrics be equally useful in democracies where the primary electoral cleavage is not region-based, i.e. spatially clustered? Answering these questions matter for generalizability of the study since in most cases, obtaining electoral survey data as fine-grained as the one used here is difficult and understanding the trade-offs is important.

Finally, a technical question. Did the authors conduct any kind of boundary sensitivity analysis for the spatial estimates? The data are organized by artificially drawn government districts, but there is good historical reason given South Korea’s electoral politics to increase the bounds to larger regional swaths, especially in Yeongnam and Honam. Because spatial clustering essentially depends on how the boundaries are defined in the estimation, this should be an important step in the analysis.

Author Response

We thank Reviewer 2 for his or her very useful comments.

Please find our detailed response in the file attached.

Round 2

Reviewer 1 Report

The authors successfully addressed all of the comments. The paper would be a good contribution to the literature.

Reviewer 2 Report

Authors have addressed the main concerns in a satisfactory manner.